

# Uncertainty in gap filling and estimating the annual sum of carbon dioxide exchange for the desert Tugai forest, Ebinur Lake Basin, Northwest China

Dexiong Teng[1,2], Xuemin He[1,2], Jingzhe Wang[1,2], Jinlong Wang[1,2] and Guanghui Lv[1,2]

[1] College of Resources and Environment Science, Xinjiang University, Urumqi, Xinjiang, China
[2] Key Laboratory of Oasis Ecology of Education Ministry, Xinjiang University, Urumqi, Xinjiang, China

## ABSTRACT

In most eddy covariance (EC) studies, carbon flux measurements have a high defect rate for a variety of reasons. Obtaining the annual sum of carbon dioxide exchange requires imputation of data gaps with high precision and accuracy. This study used five methods to fill the gaps in carbon flux data and estimate the total annual carbon dioxide exchange of the Tugai forest in the arid desert ecosystem of Ebinur Lake Basin, Northwest China. The Monte Carlo method was used to estimate the random error and bias caused by gap filling. The results revealed that (1) there was a seasonal difference in the friction velocity threshold of nighttime flux, with values in the growing season and non-growing season of 0.12 and 0.10 m/s, respectively; (2) the five gap-filling methods explained 77–84% of the data variability in the fluxes, and the random errors estimated by these methods were characterized by non-normality and leptokurtic heavy tail features, following the Laplacian (or double-exponential) distribution; (3) estimates of the annual sum of carbon dioxide exchange using the five methods at the study site in 2015 ranged from −178.25 to −155.21 g C m$^{-2}$ year$^{-1}$, indicating that the Tugai forest in the Ebinur Lake Basin is a net carbon sink. The standard deviation of the total annual carbon dioxide exchange sums estimated by the five different methods ranged from 3.15 to 19.08 g C m$^{-2}$ year$^{-1}$, with bias errors ranging from −13.69 to 14.05 g C m$^{-2}$ year$^{-1}$. This study provides a theoretical basis for the carbon dioxide exchange and carbon source/sink assessment of the Tugai forest in an arid desert ecosystem. In order to explore the functioning of the Tugai forest at this site, a greater understanding of the underlying ecological mechanisms is necessary.

## INTRODUCTION

In recent decades, with the development of sonic anemometers, the eddy covariance method has become one of the direct techniques for determining the surface-atmosphere carbon dioxide exchange (*Baldocchi, 2014*). This method consists of calculating the covariance of fluctuations in vertical wind speed and gas concentration, providing valuable

Corresponding author
Guanghui Lv,
Guanghui_xju@sina.com

data on the dynamic interactions of the surface and atmosphere. At present, more than 800 EC stations around the world are monitoring the carbon dioxide and water exchange of the surface-atmosphere. These stations are located in forests, grasslands, farmlands and other terrestrial ecosystems, concentrated in the grasslands of North America, Europe and Africa, as well as in forest ecosystems (*Baldocchi, Chu & Reichstein, 2018*; *Rebane et al., 2019*). However, there have been few consecutive observations of material fluxes in arid desert ecosystems, leading to great difficulties in understanding the role of arid desert ecosystems in the global carbon cycle.

Eddy covariance is influenced by numerous factors during the measurement of carbon dioxide ($CO_2$) flux (e.g., limited turbulence, non-stationarity, blizzards, sand, precipitation and instrument resolution), usually resulting in data loss or abnormalities. One study found that missing or abnormal data in carbon flux observations account for approximately 20–60% of the total (*Papale et al., 2006*). This poses enormous difficulties for estimating the annual sum of carbon dioxide exchange. The important step in this estimation is gap filling. To solve this problem, a number of approaches have been proposed, including non-linear regression (*Ruppert et al., 2006*), look-up tables, mean diurnal variation methods (*Falge et al., 2001*), artificial neural networks (*Braswell et al., 2005*; *Moffat et al., 2007*; *Papale & Valentini, 2003*), MDS (*Reichstein et al., 2005*), data assimilation, and Bayesian model approaches (*Gove & Hollinger, 2006*). The differences among gap-filling methods for missing data and their impacts on the estimation of carbon dioxide exchange vary temporally and spatially in urban, forest, and grassland ecosystems (*Kunwor et al., 2017*; *Liu et al., 2010*; *Moffat et al., 2007*). Establishing a standard set of gap-filling methods can enhance comparability between different sites in regional and global flux networks (*Falge et al., 2001*). Models based on the light and temperature response are widely used for filling gaps in carbon flux observations of natural vegetation (*Desai et al., 2005*). Gap-filling approaches that rely on the average temperature underestimate flux values compared to those that preserve the observed variance as a consequence of Jensen's inequality, adding further bias to the long-term estimates of carbon dioxide exchange (*Moffat et al., 2007*). The introduction of additional predictors, such as vapor pressure deficit and soil moisture content, may reduce the bias that has been found to be proportional to the percentage of gaps filled in the annual sums of carbon dioxide exchange (*Falge et al., 2001*). *Hagen et al. (2006)* utilized two regression models to fill the gaps of carbon dioxide exchange in Howland Forest, Maine, USA, concluding that the greatest uncertainty comes from model differences. The Gaussian processes (GP) model and the radial basis function (RBF) neural network have been shown to significantly improve model performance by capturing complex relationships in carbon dioxide exchange studies (*Menzer et al., 2013*; *Menzer et al., 2015*; *Schmidt, Wrzesinsky & Klemm, 2008*). There have been few studies on gap filling and the annual sum of carbon dioxide exchange in arid desert ecosystems, and uncertainty still remains in the estimation and evaluation of the annual sum of carbon dioxide exchange (*Soloway et al., 2017*). The evaluation of gap-filling uncertainty plays an important role in estimating regional carbon sink strength. The scientific comparison and evaluation of carbon flux gap-filling and the carbon budget in arid desert ecosystems are conducive to the sharing of flux data and its related research.

Arid desert ecosystems are playing an increasingly important role in the global carbon cycle (*Donohue et al., 2013*) and may even supplant tropical rainforests, affecting the interannual variation in the global carbon cycle (*Poulter et al., 2014*). Recent findings that desert regions remove carbon dioxide from the atmosphere at a magnitude of ~100 g C m$^{-2}$ year$^{-1}$ suggest that these systems may explain at least a portion of the terrestrial carbon sink that has not been fully identified nor had its mechanisms explained in the global carbon cycle (*Jasoni, Smith & Arnone, 2005*; *Ma et al., 2014*; *Stone, 2008*; *Wohlfahrt, Fenstermaker & Arnone, 2008*; *Xie et al., 2009*). Irrigated saline/alkaline arid land is a potentially large carbon sink in the terrestrial ecosystem (*Li et al., 2015*). The study of carbon flux has direct implications for the missing global carbon sink. The Ebinur Lake Basin is a typical salinized area (*Wang et al., 2019b*). The Tugai forest is the only natural forest community in the Ebinur Lake Basin, and its contribution to the carbon dioxide exchange of the region cannot be overlooked. Estimating the annual sum of the carbon dioxide exchange of the Tugai forest can provide insight into the issue of the missing carbon sink to a certain extent.

In this study, EC techniques were used to obtain continuous records of the carbon flux of the Tugai forest on the northern bank of the Aqikesu River in the Ebinur Lake Wetland National Nature Reserve, Northwest China. We evaluated the ability of five gap-filling methods to analyze random uncertainties and bias in order to provide more robust estimations of the annual sums of the carbon dioxide exchange from the Tugai forest. In addition, the impacts of these different gap-filling methods—the artificial neural network for backward propagation algorithm (ANN), the radial basis function neural network, the Gaussian processes model (GP), the regularized hierarchical linear model (HLM) with a Bayesian framework, and marginal distribution sampling (MDS)—on the annual carbon dioxide exchange in the Tugai forest were examined and the random uncertainties were studied.

## MATERIALS AND METHODS

### Site description

The study site is located in the Ebinur Lake Wetland National Nature Reserve in the Bortala Mongolian Autonomous Prefecture of the Xinjiang Uygur Autonomous Region, Northwest China (44°37′05′′–45°10′35′′N, 82°30′47′′–83°50′21′′E). The Xinjiang Ebinur Lake Wetland National Natural Protection Zone Management Bureau approved site access. With a temperate continental climate, the reserve located in the Alataw Pass (Fig. 1A) is a typical arid desert, featuring high temperatures, drought, and sufficient sunshine in summer, and dry and cold conditions in winter. With strong winds from the Alataw Mountain Pass, the loose and relatively thick Quaternary sediment provides a rich sand source for sand shifting, and the soil salinization is generally pronounced. Typical soils in the study area are gray desert soil, gray-brown desert soil, and aolian sandy soil. The azonal soil is dominated by saline soil, and supplemented by meadow soil, bog soil, and brown calcic soil (*Jin et al., 2010*; *Wang et al., 2019a*). The annual average temperature in the basin is 6–8 °C, the annual precipitation is approximately 100 mm, and the annual average sunshine is 2,800 h. The ecological environment is relatively fragile and
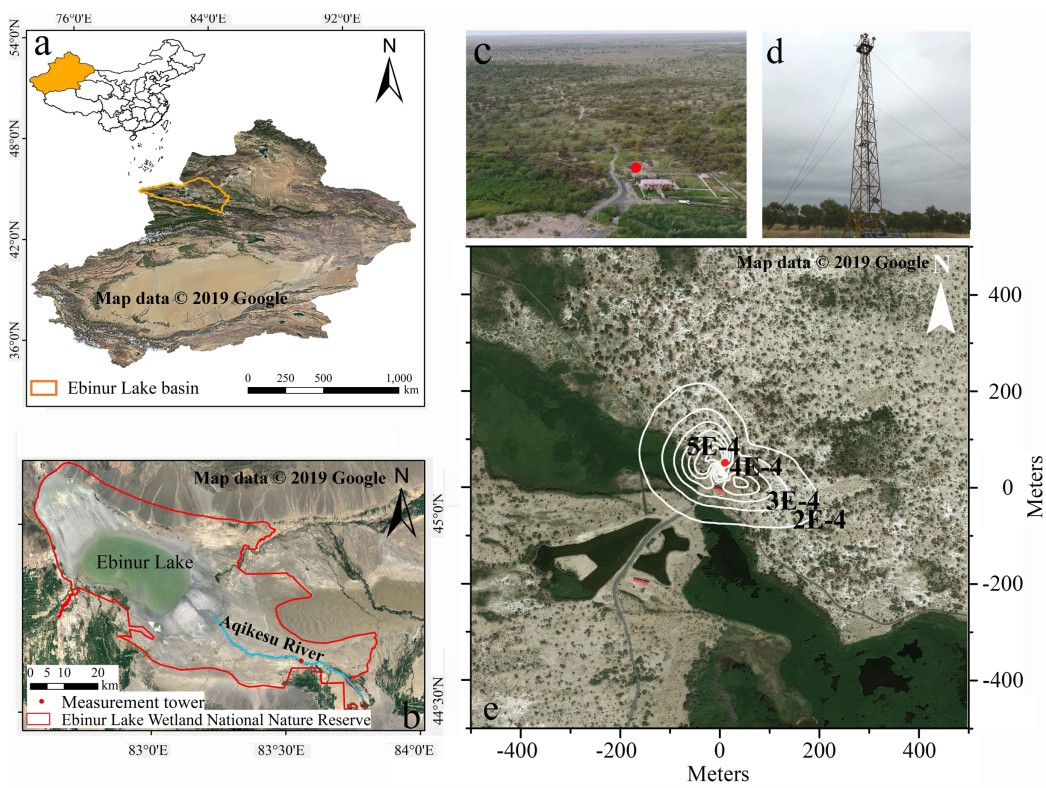

**Figure 1 Aerial photograph of the tower site.** (A) Location map of the Ebinur Lake Basin, (B) location map of the measurement tower, (C) aerial photo of the tower site in August (Photograph credit: Jinlong Wang), (D) photograph of the tower site in May (Photograph credit: Dexiong Teng), and (E) remote sensing image in June and the average footprint for the entire observation period (*Kormann & Meixner, 2001*). The measurement tower is marked with a red circle. The white lines represent isopleths of the average contributions.

consists primarily of two ecological ecosystems: the wetland ecosystem and the arid grassland-forest ecosystem (Fig. 1B). The flux observation tower stands near the Aqikesu River (44°37′4.8′′N, 83°33′59.4′′E) in the forest-desert transitional zone influenced by lakes and rivers (Fig. 1E). The dominant species of plants in this region are *Populus euphratica*, *Haloxylon ammodendron* and *Phragmites australis*, with a combined coverage of more than 60%, accompanied by a variety of shrubs, herbs and desert-specific short-lived plants (*He, Lv & Qin, 2014*, *He et al., 2015*; *Wang et al., 2019*). During the observation period, the annual average temperature at the flux tower was 9 °C, the highest temperature was 43 °C, and the lowest temperature was −26 °C. More weather conditions are shown in Fig. 2.

## Carbon flux data observation and processing

The data for this study were obtained from the Ebinur Lake Desert Ecosystem Flux Tower (2012XJ-AibiHu-OPEC) using an EC system consisting of a 3-D sonic anemometer (CSAT3; Campbell Scientific, Logan, UT, USA) and an open-path infrared gas analyzer (EC150; Campbell Scientific, Logan, UT, USA). The instruments were installed on the southwest side of the tower above the forest canopy at a height of 15.0 m (~4.0 m above

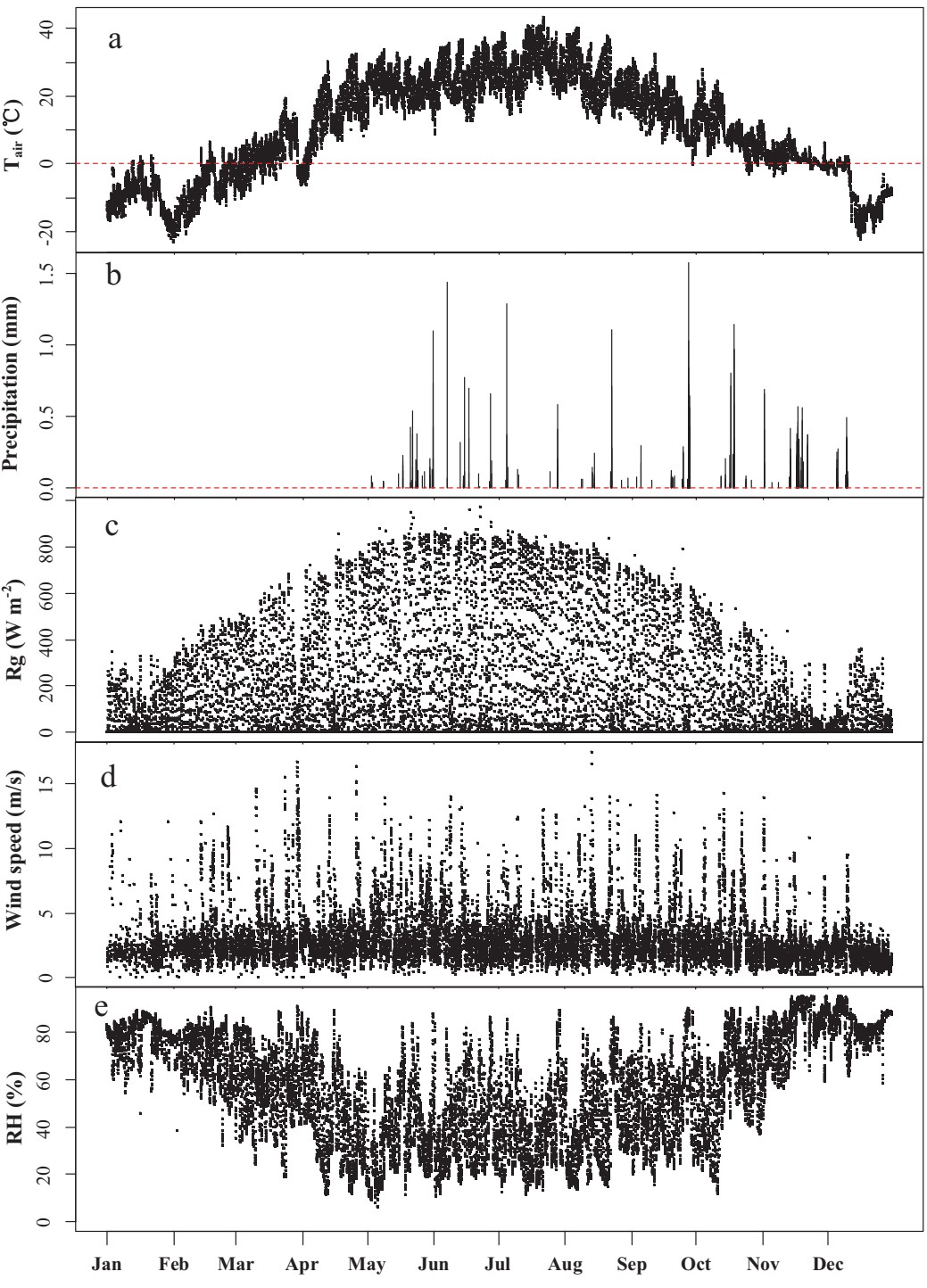

**Figure 2 Meteorological observations during the year of measurement.** Half-hour averaged values of (A) air temperature ($T_{air}$), (B) precipitation, (C) global radiation (Rg), (D) wind speed and (E) relative humidity (RH) during the observation period.

the mean canopy height). Total $CO_2$ fluxes were measured from January 1, 2015 to December 31, 2015. Using EddyPro 6.2.1 software, 30 min averaged values were calculated, resulting in 17,520 observations (Table 1). Webb, Pearman and Leuning (WPL) correction

**Table 1 The data source of the driving factors and their corresponding pre-processing.**

| Variable | Name | Filled? | Pre-processing |
|---|---|---|---|
| Daytime | $hour_N$ | No | Cos conversion to daily time |
| Daytime | $hour_D$ | No | Sin conversion to daily time |
| Daytime | Night | No | Calculated the sunrise and sunset times at the observation area location and marked the data from post-sunset to pre-sunrise as nighttime |
| Daytime | Day | No | Calculated the sunrise and sunset time at the observation area location and marked the data from post-sunrise to pre-sunset as daytime |
| Season | $Season_G$ | No | Marked the data from May–September as the growing season |
| Season | $Season_N$ | No | Marked data from January–April and October–December as the non-growing season |
| Vegetation index | NDVI | Yes | NDVI from 250 m resolution MODIS/Aqua remote sensing image for 16 days |
| Air temperature | $T_{air}$ | Yes | Measured at a height of 15 m on the flux tower |
| Soil temperature | $T_{soil}$ | Yes | Measured at a soil depth of 5 cm under the flux tower |
| Water vapor partial pressure | E | Yes | Measured at a height of 15 m on the flux tower |
| Heating degree days | hdd | Yes | When $T_{air} < 18\ °C$, equal to $18\ °C—T_{air}$; otherwise, equal to 0 |
| Incident radiation | Rd | Yes | The sum of incident shortwave radiation and incident longwave radiation measured by NR01, Campbell Scientific, USA |
| Scattered radiation | Ru | Yes | The sum of scattered shortwave radiation and scattered longwave radiation measured by NR01, Campbell Scientific, USA |
| Water vapor pressure deficit | VPD | Yes | Measured at a height of 15 m on the flux tower |
| Relative humidity | Rh | Yes | Measured at a height of 15 m on the flux tower |

was used during the data pre-processing (*Webb, Pearman & Leuning, 1980*). Due to extremely low temperatures and instrument failure 4,858 (approximately 27.73%) of the data observations were missing. Based on the stationarity test (*Foken & Wichura, 1996*; *Mauder et al., 2013*), the observations were divided into three quality classes (0, 1 and 2). Only high-quality data were selected to establish the model, and 5,784 (approximately 33.01%) of the data observations with a rank other than zero were excluded. Based on changes in surface vegetation, the year can be divided into the growing season (May–September) (*Munir et al., 2015*) and the non-growing season (January–April and October–December). This study divided the data into daytime and nighttime periods based on the sunrise and sunset times at the location of the flux tower. The observation time of the daytime data extended from sunrise to sunset, and the observation time of the nighttime data extended from sunset to sunrise. There were 8,924 daytime observations, accounting for 50.94% of the total and 8,596 nighttime observations, accounting for 49.06%. The total count of null and low-quality observations was 6,735, accounting for 38.44%. Thus, there was a large amount of missing data, and data gap-filling faced enormous challenges.

Corresponding to the 30 min carbon dioxide flux ($F_C$) observation data, eight explanatory variables, including soil temperature, air temperature, relative humidity, incident radiation, scattered radiation, saturated vapor pressure deficit, normalized difference vegetation index (NDVI), and water vapor partial pressure, were selected as the driving factors for the gap-filling models. Partial explanatory variables needed to be converted (Table 1). The selected meteorological variables were normalized by mapping all explanatory variables to (0, 1) via linear transformation and using the 2 variables of growing and non-growing to describe the season, both of which are binary (0 or 1). Based on the sunrise and sunset times, the data were classified into 2 types, daytime and nighttime, which were represented by binary variables. The observation data were divided into 3 parts: high-quality data used for training and testing models; data used to calculate the annual exchange of carbon flux; and gaps.

## Net ecosystem carbon dioxide exchange

The net ecosystem carbon dioxide exchange (NEE) between the ecosystem and the atmosphere can be estimated as the sum of the vertical EC and the storage term:

$$NEE = F_C + F_S \tag{1}$$

The low density of the Tugai forest canopies, which is favorable to eddy penetration, can reduce the importance and frequency of storage events. The lower the vegetation height, the less storage of $CO_2$. In addition, the effect of the storage terms on the carbon uptake characteristics over long-term time scales is not significant since the cumulative value of the storage terms in the long-term is approximately equal to 0 (*Massman & Lee, 2002*). In this study, the storage terms were considered to be negligible compared with the terms associated with EC fluctuation.

## Night friction velocity threshold

The friction velocity $u^*$ is intended to capture the turbulence characteristics in the vicinity of the surface. In a general way, $u^*$ can be expressed as a fraction of the average wind speed $u$:

$$u^* \approx \frac{u}{\Pi} \tag{2}$$

where the parameter $\Pi$ is the roughness index. We used standardized major axis (SMA) regression to compare the roughness difference between the growing season and the non-growing season (*Warton et al., 2012*).

Due to the relatively stable atmospheric stratification at night, the carbon flux observed using EC techniques cannot truly represent the surface-atmosphere carbon dioxide exchange. Nighttime data processing was an important step in the flux data processing. The uncertainty of the annual sum of ecosystem carbon exchange caused by different treatment methods can reach 150 g C m$^{-2}$ year$^{-1}$ (*Williams et al., 2009*). The bias due to the data processing methods in this study area, as well as the estimated bias from the gap-filling techniques, may be greater. To correct selective systematic errors, the international flux community usually excludes nighttime observations below the friction

velocity threshold in order to ensure that EC techniques are measured under strong turbulence conditions (*Aubinet & Feigenwinter, 2010*). Currently, despite theoretical and practical deficiencies, the empirical and effective method for processing nighttime negative flux is filtered by the friction velocity threshold (*Aubinet & Feigenwinter, 2010*). The friction velocity threshold can be determined by testing the relationship between the flux and the friction velocity, since, in theory, carbon dioxide flux and friction velocity are independent. When the friction velocity is lower than this threshold, $F_C$ decreases as the friction velocity decreases. The nighttime flux data of the growing and non-growing seasons were sorted based on the increment of $u^*$, and the friction velocity thresholds for the growing and non-growing seasons were then determined. A total of 45 levels were set, and the thresholds were derived by statistically comparing (using the two-sample Wilcoxon test) the flux average at each $u^*$ level with the flux average at the next higher $u^*$ level.

## Detection of abnormal data points

The technique of using EC to observe carbon flux is susceptible to environmental factors such as rapid changes in turbulence, as well as the instrumentation itself. The 30 min data obtained from processing the observed raw data contained a large number of gaps and abnormal values, which deviated from the range of normal data variation (*Li et al., 2005*). Given the relatively clear daily variation of the carbon flux data, this study divided the carbon flux data into 48 parts according to the time of day and defined any carbon flux data beyond the upper and lower limits as abnormal points. For outliers in the continuous data, this study used 7 days as the window to determine the difference between the median of three consecutive data points ($D_i$). The specific algorithm (*Papale et al., 2006*) was as follows:

$$D_i = (F_{Ci} - F_{Ci-1}) - (F_{Ci+1} - F_{Ci}) \tag{3}$$

where $F_C$ is the carbon dioxide flux, and $i$ is the ordinal number of the observation data for 30 min. When $D_i$ satisfies the relationship

$$\text{Md} + (z * \text{MAD}/0.6745) > D_i > \text{Md} - (z * \text{MAD}/0.6745) \tag{4}$$

the corresponding $i$th observation data point is normal; otherwise, it is defined as an "abnormal point" and is eliminated (Daytime and nighttime data were processed separately). Md is the median of the $F_C$ differences between all sets of two adjacent points. MAD = median ($|D_i - \text{Md}|$) and is the median of all $|D_i - \text{Md}|$ for a 7-day window. In this study, the sensitivity coefficient $z$ of the artificially defined outlier identification was set to 5.5.

## Gap-filling methods

Given the actual conditions of a site, selecting the appropriate flux data processing method and reducing the uncertainty of the flux data estimation are important issues that need to be understood and analyzed by the flux community. The effectiveness of different methods for filling gaps in carbon dioxide flux ($F_C$) data has been investigated. This study used

ANN, RBF, GP, HLM and MDS to fill the gaps. The high-quality data were randomly divided into two parts, 80% of which were used to train the model and 20% to test the model. In order to avoid overfitting, four different methods (ANN, GP, HLM and RBF) were used for model training before the best model for each method was selected and used for prediction. In the data analysis, the neuralnet, RSNNS, laGP and INLA packages (R language) were used to train and test ANN, RBF, GP and HLM, respectively. The gap-filling model was evaluated in terms of three statistical factors: the coefficient of determination ($R^2$), root mean square error (RMSE), and bias error (BE).

### ANN and RBF

For ANN training, the input data were passed through the nodes using weighted connections, the error between the predicted and actual output values was calculated by backpropagation, and the weight was adjusted to minimize the error (*Braswell et al., 2005*; *Papale & Valentini, 2003*). The training data were pre-sampled to ensure equal coverage under different conditions, and fuzzy values were used to represent additional information such as time (*Moffat et al., 2007*; *Papale & Valentini, 2003*). By testing the structure of the ANN, the network was built to be as simple as possible, and overfitting was avoided.

During the model training process, with the exception of the input and output layers, only one hidden layer was needed to obtain sufficient results; additional hidden layers did not improve model performance. In the output layer, a linear transfer function was used. Both the ANN and RBF were built using the input variables listed in Table 1, as well as the 30 min data points of $F_C$, with the same parameters presented by *Schmidt, Wrzesinsky & Klemm (2008)*. The number of neurons in the ANN hidden layer was 5, and the number of neurons in the RBF hidden layer was 181.

### Regularized hierarchical linear regression model with Bayesian framework

Since signals may be less pronounced than random errors, overfitting is a potential problem in regression models in which parameters estimate noise rather than the underlying signal. In earlier works, hierarchical modeling produced more accurate predictions than regular regression and also tended to reduce overfitting in the presence of a hierarchical data structure (*Gelman, 2006*). There have been some detailed examples highlighting uses of Bayesian and hierarchical Bayesian methods in plant physiology and ecosystem ecology (*Ogle & Barber, 2008*). Hierarchical modeling has been proposed to predict monthly streamflow (*Lima & Lall, 2010*) and extreme flooding (*Lima et al., 2016*). This study used varying intercepts of hierarchical modeling for the carbon flux data with a temporally hierarchical structure. The form of the model was as follows:

$$y_{ij} \sim N\left(\alpha_j + \beta x_{ij},\ \sigma_y^2\right),\ \text{for } i = 1, \ldots,\ 365;\ j = 1, \ldots, 48 \tag{5}$$

$$\alpha_j \sim N\left(\gamma_j,\ \sigma_{\alpha_j}^2\right) \tag{6}$$

$$\beta | \sigma_\beta^2 \sim N\left(0,\ \sigma_\beta^2 I\right) \tag{7}$$

where $y_{ij}$ is $F_C$ at time $j$ within Julian Day $i$, and $x_{ij}$ is a vector for the environmental measurement (Table 1) corresponding to $F_C$. The hyperparameters of the prior distribution of $\alpha_j$ are $\gamma_j$ and $\sigma^2_{\alpha_j}$. In previous studies, the prior ridge has been chosen for shrinking coefficients due to its weaker penalization of large coefficients (*Fahrmeir, Kneib & Konrath, 2010*). Therefore, we assumed that $\beta|\sigma^2_\beta \sim N\left(0,\ \sigma^2_\beta I\right)$.

### *Gaussian processes*

As nonlinear regression tools, GP have nonparametric flexibility and can be used to fit arbitrary functions or surfaces (*Azmy et al., 2009*). Therefore, the Gaussian process is often used to replace the supervised neural network in nonlinear regression. The GP can be described by a mean function $m(x)$ (a vector) and a covariance function $k(x, x')$ (a matrix):

$$f(x) = GP(m(x),\ k(x,\ x'))\qquad(8)$$

where $f(x)$ is a random variable. In this study, $f(x)$ was the modeled $F_C$. This assumes that the mean function $m(x)$ is equal to 0 since our data were standard normalized, and $k(x,\ x')$ is a function of the Euclidean distance between the predictors $x$ and $x'$. The squared exponential covariance function was chosen with automatic relevance determination (*Bishop, 2006*):

$$\text{cov}(f(x),\ f(x')) = k(x,\ x') = \sigma^2_f \exp\left(-\frac{1}{2}\sum_{i=1}^{D}\frac{\|x_i - x'_i\|}{\sigma^2_i}\right)\qquad(9)$$

where $\sigma^2_f$ is a hyperparameter for the amplitude variance. The width hyperparameter $\sigma^2_i$ is separate for each input variable $x_i$.

## Characteristics of random uncertainty

The correlation between the predicted carbon flux and its corresponding residual is represented by Kendall's tau ($\tau$) coefficient. If $\tau$ is close to 0, the posterior residual does not contain structural error from the model. Conversely, if $\tau$ is close to 1, the predicted value should contain the model structure error of the influence, and such interpretation information may be passed to the posterior residual (*Wang, Riley & Collins, 2015*).

The relationship between the standard deviation of the random uncertainty of carbon flux observations and the magnitude of the observed values is commonly expressed as follows:

$$\sigma(\varepsilon) = a\ +\ b|F_C|.\qquad(10)$$

The intercept term $a$ varies from site to site, with a typical range of 0.9–3.5 µmol m$^{-2}$ s$^{-1}$ (*Richardson et al., 2008*). The larger the value of $a$, the greater the uncertainty. The range of the slope $b$ is usually 0.1–0.2; the smaller the value of $b$, the closer the probability distribution of the uncertainty to white noise.

## Uncertainty in the cumulative carbon dioxide exchange

To determine the random error of carbon flux and the error caused by interpolation, this study used the bootstrap method to estimate the deviation of the annual sum of ecosystem carbon exchange and its 95% confidence interval. Estimating the RMSE, variance, and

bias of the annual sum of carbon exchange required a complete carbon flux time series (1,000 bootstrap sequences). The variance of the total annual carbon exchange can be directly estimated, but due to vacancies in the carbon flux data, the deviation of the total annual carbon exchange cannot be directly estimated. To estimate the (potential) bias in carbon flux vacancy data, we used the bootstrap method to simulate and analyze the bias of high-quality modeling data in order to extend it to an annual scale (*Richardson et al., 2006*; *Wang, Riley & Collins, 2015*). The estimation steps used to determine the bias of the annual sum of carbon exchange were as follows:

1. The vector $\hat{\varepsilon}$, which is the estimation of the random error $\varepsilon$, was estimated by subtracting the observed $F_C$ from the $F_C$ estimated by the models.
2. All data (observed $F_C$ with gaps as well as modeled $F_C$) were divided in terms of the growing season, daytime, non-growing season, and nighttime. The daytime data of the growing season were divided according to the incident radiation intensity (limited to 400 W·m$^{-2}$), resulting in 6 parts.
3. $\hat{\varepsilon}$ was repeatedly sampled in order to obtain a new complete bias sequence (a total of 17,520 repeated extractions per year).
4. A cumulated summation of the new complete bias sequence was calculated.
5. Steps (3) and (4) were repeated 1,000 times.
6. The empirical distribution of the 1,000 biases of the annual sum of ecosystem carbon exchange, along with the 95% confidence interval of the annual sum of carbon exchange, were estimated.

# RESULTS

## Carbon dioxide flux data characteristics

To model the high-quality carbon flux data, the 30 min carbon flux data with high signal quality were screened. It can be seen that the flux observation points exhibit more obvious carbon uptake characteristics during the growing season (Fig. 3). To extract the nighttime flux error, the nighttime friction velocity thresholds for the growing and non-growing seasons were estimated, and the data on friction velocities less than the threshold were screened and eliminated. This amounted to approximately 2.4% of the data (415 data points). After processing for outliers and nighttime friction velocity threshold screening, approximately 35.9% (or 6,293 observations) of the high-quality data remained, which was then used to build and test the model (Table 2). Among these measurements, there were 2,892 valid daytime observations (~16.5%) and 1,032 valid nighttime observations during the growing season (~5.9%). There were 1,583 valid daytime observations (~9%) and 786 (~4.5%) valid nighttime observations during the non-growing season.

Within the boxes, the horizontal bars indicate medians, while the tops and bottoms of the boxes illustrate the 75th and 25th quartiles, respectively. Small circles represent outliers in the observations.

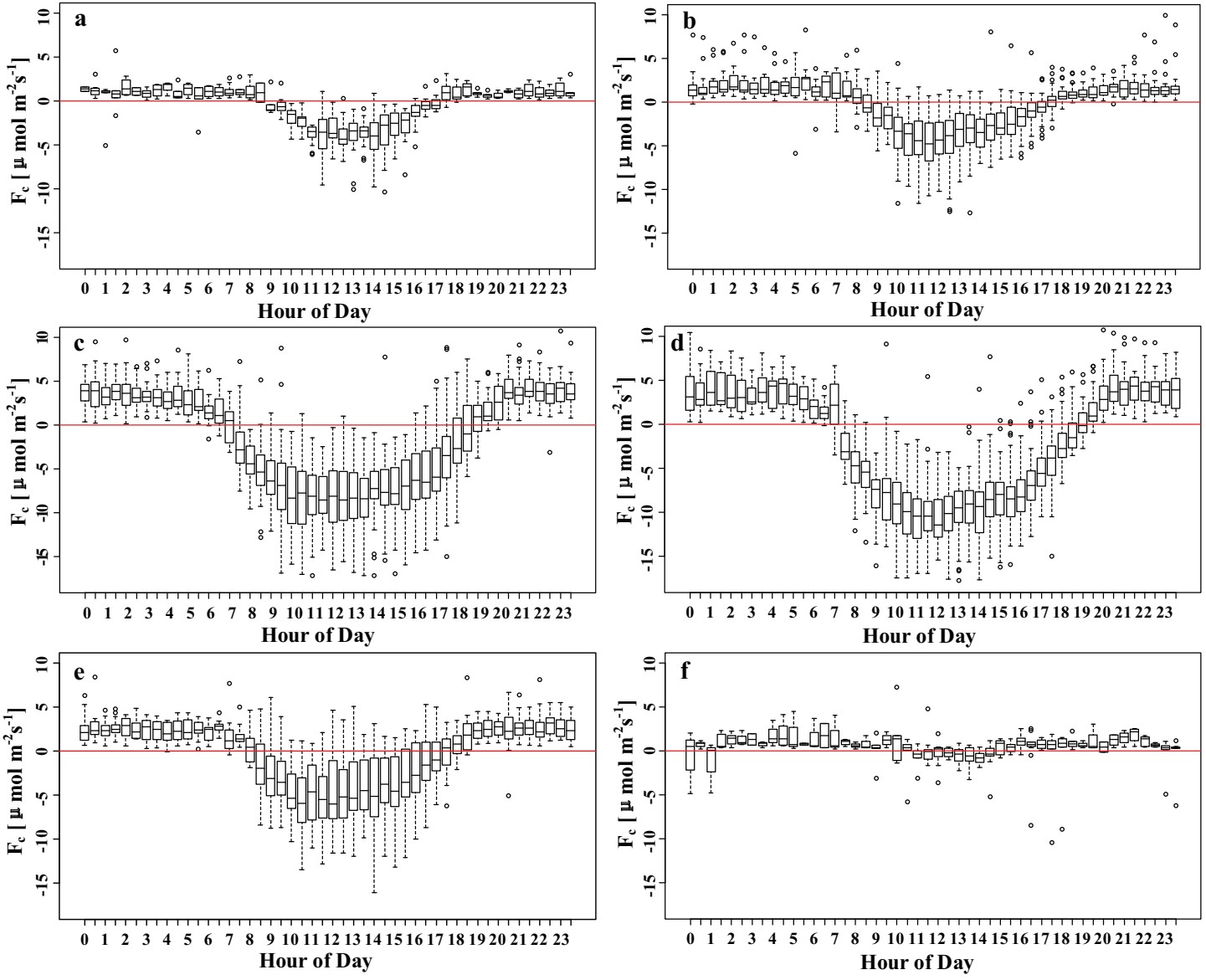

**Figure 3 Mean diurnal courses of high-quality 30 min $F_C$ measurements averaged over 2 month periods after filtering: (A) January–February, (B) March–April, (C) May–June, (D) July–August, (E) September–October and (F) November–December.** Within the boxes, the horizontal bars indicate medians, while the tops and bottoms of the boxes illustrate the 75th and 25th quartiles, respectively. Small circles represent outliers in the observations.

This study defined the length of time in which data were continuously missing as the missing time window. A histogram plot was used to display the distribution of the missing time window (Fig. 4). The average length of the missing time window for the flux data was 5.8 h; a total of 24 missing time windows were longer than 24 h, mainly concentrated in the non-growing season; the longest missing time window was 30 days (Fig. 5). It can be seen that wind speed and friction velocity $u^*$ differed significantly between the growing and non-growing seasons (Fig. 6). The details of the relationship between $u^*$ and wind speed using SMA regression are presented in Table 3; Fig. 6C.

**Table 2 Quality control procedures for the 30 min flux data.**

| Processing | The number of outliers | | Description |
|---|---|---|---|
| | Growing season | Non-growing season | |
| Quality control | 856 | 1,021 | Cleaning carbon flux data based on QC flags and removing values while QC flags equal to 2 |
| Distributional cleaning | 679 | 464 | Cleaning carbon flux data based on flux distribution for each hour of the day |
| Filtering | 62 | 26 | Detecting and removing outliers in the continuous carbon flux data |
| $u^*$ screening | 216 | 192 | Cleaning carbon flux data based on friction vorticity threshold |

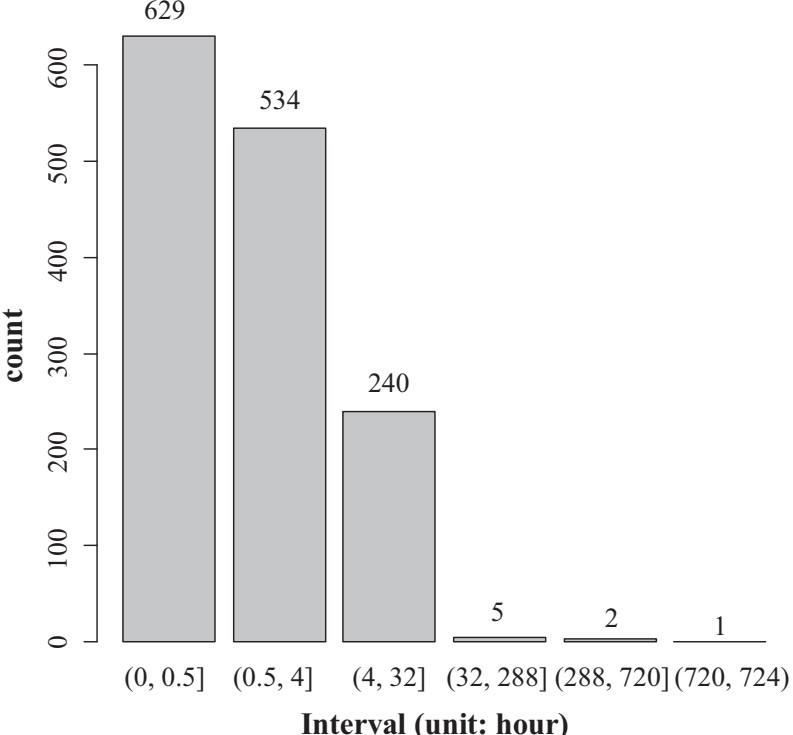

**Figure 4 Histogram of the data gap distribution.**

The slopes of the SMA regression for the growing and non-growing seasons were not equal, indicating that the roughness index varied considerably with season.

Red, orange, and gray represent Qc = 0, Qc = 1 and Qc = 2, respectively.

The nighttime friction velocity threshold during the growing season was 0.12 m/s, and the corresponding value for the non-growing season was 0.10 m/s. Detailed information concerning the quality filtering procedures is listed in Table 2. A total of 42 negative $F_C$ values were observed at night in this study, accounting for 1.9% of the nighttime data. After removal by the friction velocity threshold and other quality filtering procedures, the number of nighttime negative values was reduced to 16 (~0.7%).

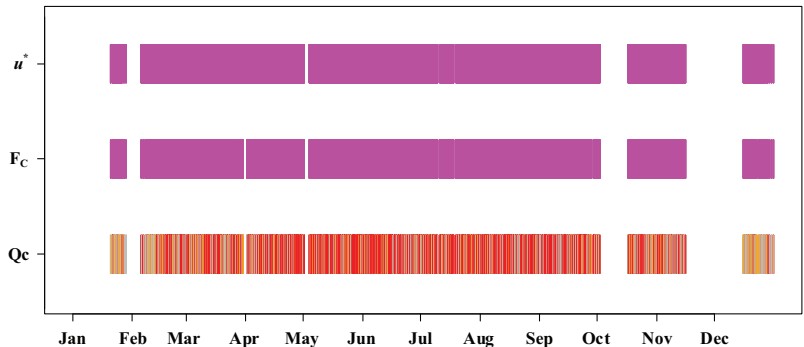

**Figure 5 Time of $F_C$ and friction velocity $u^*$ observation times over the study year.** Red, orange, and gray represent Qc = 0, Qc = 1, and Qc = 2, respectively.

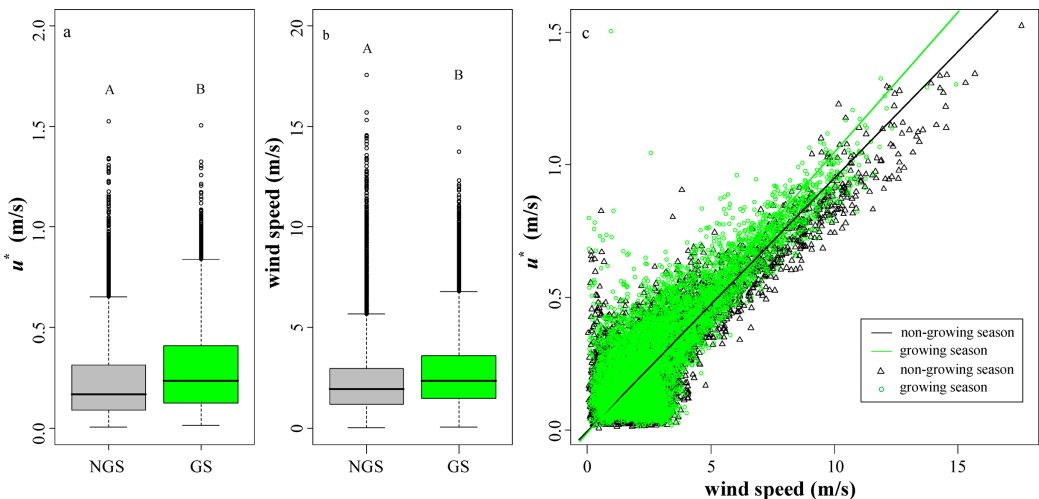

**Figure 6 Values of nighttime friction velocity $u^*$ and wind speed.** (A) Nighttime friction velocity $u^*$. (B) Nighttime wind speed. (C) The relationship between friction velocity and wind speed for all data was grouped by season. Green and gray represent the growing and non-growing seasons, respectively. Columns with different capital letters are significantly different ($P < 0.05$) according to Fisher's least significant difference test.

**Table 3 Standardized major axis estimation and statistical inference.**

| Parameter | Non-growing season | | | Growing season | | |
|---|---|---|---|---|---|---|
| | Estimate | Lower (2.5%) | Upper (97.5%) | Estimate | Lower (2.5%) | Upper (97.5%) |
| Elevation | −4.039e−3 | −8.112e−3 | 3.399e−5 | −0.011 | −0.015 | −0.006 |
| Slope | 0.095 | 0.094 | 0.097 | 0.106 | 0.104 | 0.107 |
| $H_0$: variables uncorrelated | | | | | | |
| $P$-value | <2.22e−16 | | | <2.22e−16 | | |
| $R^2$ | 0.730 | | | 0.706 | | |
| $H_0$: slopes are equal | | | | | | |
| $P$-value | <2.22e−16 | | | | | |
| Likelihood ratio statistic: 129.8 with 1 degrees of freedom | | | | | | |

## Results and residual analysis of gap-filling methods

The results of the ANN, GP, HLM, RB, and MDS methods are illustrated in Fig. 7. The test set $R^2$ obtained by these different methods ranged from 0.77 to 0.84 and the RMSE ranged from 2.04 to 2.50 µmol m$^{-2}$ s$^{-1}$. There were differences in the $R^2$ and RMSE obtained by these methods. The GP and MDS methods represented the full range of scatter in the observations, while the others all exhibited the upper and lower limits (Fig. 7). The training set fit curve overlapped with the test set fit curve, and they were equally close to the 1:1 line, indicating that there was no overfitting phenomenon in these methods. The gap-filled $F_C$ for the five methods exhibited a relatively obvious diurnal variation (Fig. 8).

When model error is quantified, the model residuals can quantify the random uncertainty. In this study, it was assumed that model error was negligible. Thus, the model residuals could be attributed almost entirely to random error. The empirical distribution of the model residuals and the fitted curves (Fig. 9) can describe the uncertainty of the random error and largely reflects the nature of the carbon flux observations. The model residual distributions obtained by the five methods exhibited characteristics of sharp peaks and thick tails, indicating non-normality. Comparing the kurtosis for the empirical distributions of the residuals obtained by these methods (Fig. 9), it was found that GP > MDS > ANN > HLM > RBF. Additionally, the empirical distribution of these residuals was discovered to approximate the Laplace (or double-exponential) distribution by the Wilcoxon rank sum test ($p > 0.05$).

Comparing the correlations between the carbon flux observations and the corresponding residuals obtained by the five different methods (Table 4), it was found that during the day in the non-growing season, Rd ≤ 400 W m$^{-2}$ showed a significant weak positive correlation; the same was true at night in the non-growing season. Additionally, it was shown that the random uncertainty during the non-growing season increased with increasing carbon flux. The correlations between the flux prediction values and the residuals obtained by the five methods differed during the growing season. In the growing season, the GP and MDS methods each exhibited a significant positive correlation, while the other methods showed either a relatively weak correlation (τ close to 0) or no significant correlation.

The standard deviation of the random flux measurement error varied linearly with the magnitude of the $F_C$. Results are summarized in Table 5. The range of the intercept term $a$ of the flux observation station in this study was 1.3209–1.5550 µmol m$^{-2}$ s$^{-1}$, which is within the typical range. The largest values came from the MDS method; the smallest from the GP. The uncertainties of the carbon flux gap-filling from these five methods were ranked as follows: GP < ANN < RBF < HLM < MDS. The slope $b$ ranged from 0.0579 to 0.1092, which was slightly less than the typical range. The smallest slope came from the HLM; the largest from the MDS. The posterior residual of the carbon flux gap-filling from the HLM method was closer to white noise than that from the MDS method. The carbon flux data gaps were filled by the different methods. The probability distributions of the measured random errors varied with changes in the

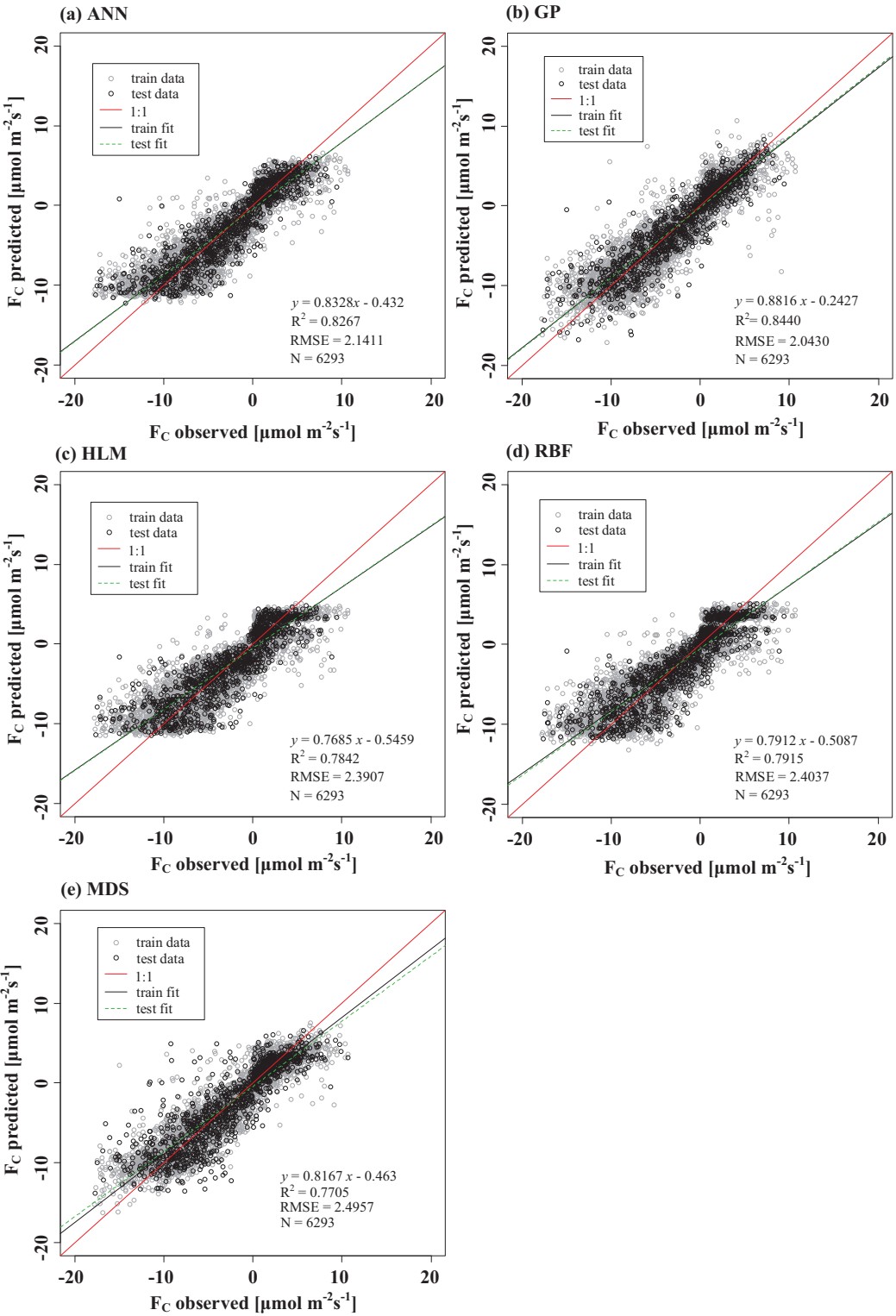

**Figure 7 Training and test performances of the models for the dataset with 6,293 points.** (A) ANN, (B) GP, (C) HLM, (D) RBF and (E) MDS. For comparison, the plotted lines consist of the training fit (solid black line), the test fit (green dotted line), and the 1:1 line (solid red). Units of RMSE and BE are $\mu$mol m$^{-2}$ s$^{-1}$.

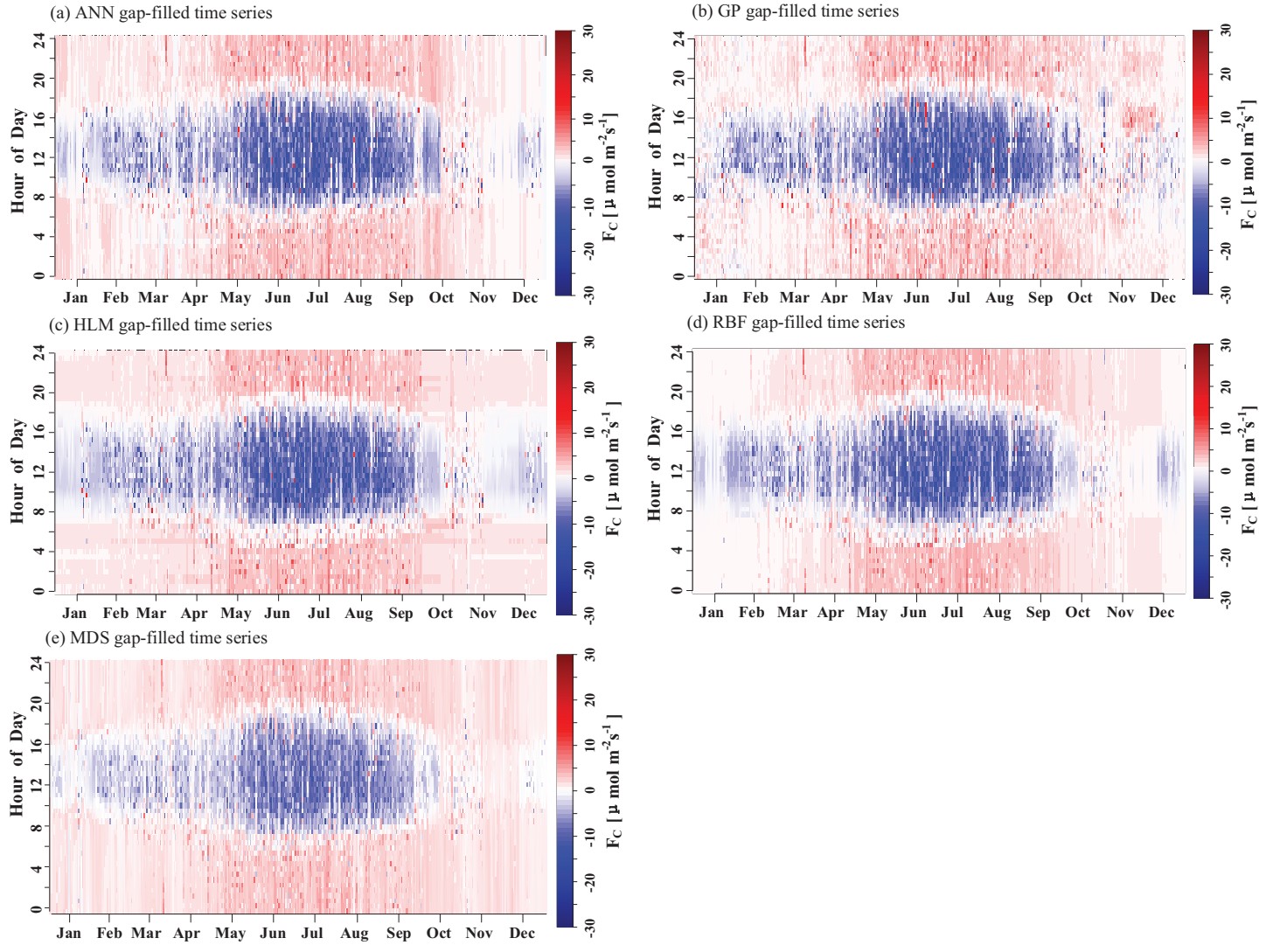

**Figure 8 Fingerprints of the complete $F_C$ gap-filled by the (A) ANN, (B) GP, (C) HLM, (D) RBF and (E) MDS methods.**

different gap-filling methods; the gap-filling methods also affected the estimation of the annual sum of carbon dioxide exchange.

## Annual sum of carbon dioxide exchange

Utilizing the gap-filling from the five methods, the complete carbon flux data were obtained. As shown in Table 6, the gap-filled annual $F_C$ sums from MC simulations were estimated to be $-159.91 \pm 18.35$ g C m$^{-2}$ year$^{-1}$ for ANN, $-178.25 \pm 8.01$ g C m$^{-2}$ year$^{-1}$ for GP, $-167.16 \pm 19.08$ g C m$^{-2}$ year$^{-1}$ for RBF, $-156.95 \pm 3.15$ g C m$^{-2}$ year$^{-1}$ for HLM, and $-155.21 \pm 5.16$ g C m$^{-2}$ year$^{-1}$ for MDS. The $CO_2$ exchange sums from MC simulations for the growing and non-growing seasons are listed in Table 7. Given that five different gap-filling models were used, it was definitely necessary to correct the model error. Based on the chosen fusion method (*Kuncheva, 2002*), it was assumed that the

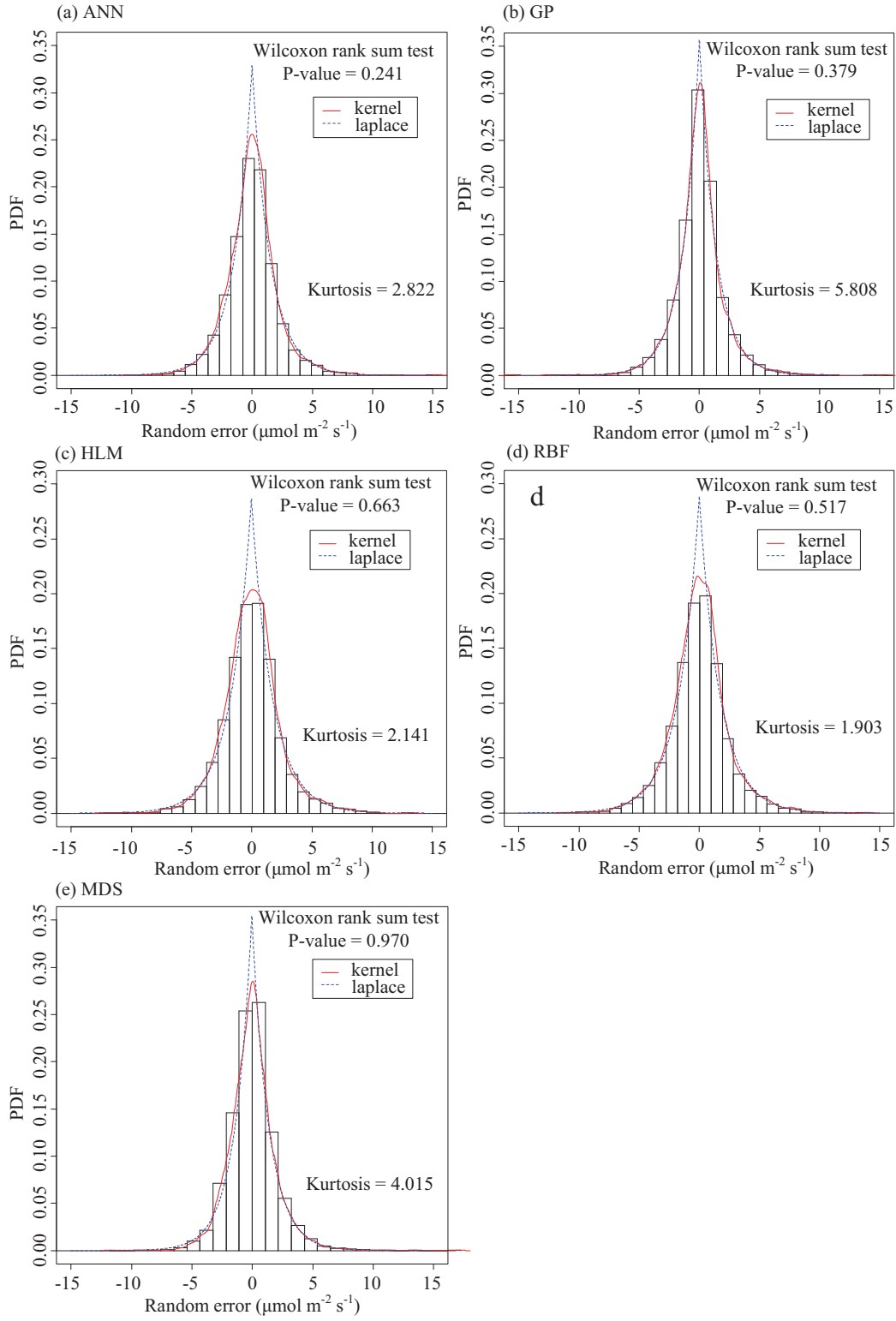

**Figure 9 Empirical distributions of the residuals predicted by the five gap-filling methods and their corresponding Laplacian distributions.** (A) ANN, (B) GP, (C) HLM, (D) RBF and (E) MDS.

**Table 4 Kendall's tau ($\tau$) coefficient between the five different methods of carbon flux prediction and the corresponding residuals.**

| Method | Non-growing season | | | Growing season | | |
|---|---|---|---|---|---|---|
| | $DAY_{Rd \leq 400}$ | $DAY_{Rd > 400}$ | NIGHT | $DAY_{Rd \leq 400}$ | $DAY_{Rd > 400}$ | NIGHT |
| GP | 0.148** | 0.171** | 0.221** | 0.077** | 0.145** | 0.118** |
| RBF | 0.048* | 0.057 | 0.202** | −0.290** | 0.082** | 0.005 |
| ANN | 0.058** | 0.005 | 0.134** | 0.009 | −0.009 | −0.059** |
| HLM | 0.079** | 0.014 | 0.192** | −0.175** | 0.033* | −0.060** |
| MDS | 0.070*** | 0.071** | 0.074** | 0.182*** | 0.075*** | 0.012 |

Notes:
* Indicates $0.05 < P$-value $< 0.1$.
** Indicates $0.01 < P$-value $< 0.05$.
*** Indicates $p$-value $< 0.01$.

**Table 5 Relationship between the magnitude of carbon flux and the standard deviation of random uncertainty.**

| Method | Intercept ($a$) | Slope ($b$) | $R^2$ |
|---|---|---|---|
| HLM | 1.5477*** | 0.0579*** | 0.3191 |
| ANN | 1.4001*** | 0.0608*** | 0.3047 |
| GP | 1.3290*** | 0.1082*** | 0.5906 |
| RBF | 1.5208*** | 0.0777*** | 0.4190 |
| MDS | 1.5550*** | 0.1092*** | 0.4513 |

Notes:
*** indicates $p$-value $< 0.01$.
$R^2$ indicates the coefficient of determination.

**Table 6 Annual sum and deviation of the carbon dioxide exchange estimated by the different methods.**

| Gap-filling method | Annual sum of $CO_2$ exchange | | Bias of Annual sum of $CO_2$ exchange | | |
|---|---|---|---|---|---|
| | Mean | SD | Mean | Lower (2.5%) | Upper (97.5%) |
| ANN | −159.91 | 18.35 | 0.84 | −9.55 | 10.73 |
| GP | −178.25 | 8.01 | 11.22 | 1.37 | 21.71 |
| RBF | −167.16 | 19.08 | 14.05 | 2.63 | 25.89 |
| HLM | −156.95 | 3.15 | −2.64 | −14.65 | 9.55 |
| MDS | −155.21 | 5.16 | −13.69 | −23.70 | −4.20 |

mean of the five methods was close to the true value of the annual sum of carbon dioxide exchange. The annual sum of $CO_2$ exchange in 2015 at the observed site was estimated to be $-163.50 \pm 9.43$ g C m$^{-2}$ year$^{-1}$, indicating that the desert Tugai forest in the Ebinur Lake Basin is a net carbon sink on an annual basis.

The averages of the annual sum of $CO_2$ exchange obtained by the five methods ranged from $-178.25$ to $-155.21$ g C m$^{-2}$ year$^{-1}$, and the standard deviation ranged from 3.15 to 19.08 g C m$^{-2}$ year$^{-1}$ (Table 6). The smallest (in terms of absolute value) came from the MDS method and the largest from the RBF. The average bias of the annual sum of ecosystem carbon exchange for the five methods ranged from $-13.69$ to 14.05 g C m$^{-2}$ year$^{-1}$.

**Table 7 Seasonal sum of carbon dioxide exchange estimated by the different methods.**

| Method | Non-growing season | | | Growing season | | |
|--------|------|--------------|---------------|------|--------------|---------------|
| | Mean | Lower (2.5%) | Upper (97.5%) | Mean | Lower (2.5%) | Upper (97.5%) |
| GP | 39.788 | 33.877 | 46.264 | −220.245 | −227.814 | −212.417 |
| RBF | 68.292 | 62.380 | 74.767 | −232.510 | −240.080 | −224.682 |
| ANN | 56.000 | 50.088 | 62.476 | −216.624 | −224.194 | −208.796 |
| HLM | 61.033 | 55.121 | 67.508 | −227.859 | −235.428 | −220.031 |
| MDS | 74.954 | 69.042 | 81.429 | −229.959 | −237.529 | −222.131 |

**Figure 10 Cumulative carbon dioxide exchange obtained by the different methods.** The data sources were composed of 3 parts, which were high-quality modeling data, data used to calculate annual sum of carbon dioxide exchange, and predictions from the different gap-filling methods.

The HLM and MDS tended to underestimate, while the other methods tended to overestimate (Table 6). The smallest mean bias of the annual sum of carbon exchange came from the ANN; the largest was from the RBF. The GP had the smallest 95% confidence interval for the bias of the annual sum of carbon dioxide exchange, while the HLM had the largest. Comparing the carbon accumulation losses of ecosystems estimated by the different methods (Fig. 10), it was found that these five methods exhibited a consistent performance during the growing season and large differences during the non-growing season.

The data sources were composed of three parts, which were high-quality modeling data, data used to calculate annual sum of carbon dioxide exchange, and predictions from the different gap-filling methods.

# DISCUSSION

## Differences in annual sums of $CO_2$ exchange among gap-filling methods for the Tugai forest

The annual sum of carbon dioxide exchange from the Tugai forest in our study was estimated to range from −178.25 to −155.21 g C m$^{-2}$ year$^{-1}$. In a previous study, a total

annual carbon exchange ranging from −155 to −55 g C m$^{-2}$ year$^{-1}$ for oak/grass savanna from 2000 to 2006 was obtained (*Ma et al., 2007*), slightly less than the estimated value in this investigation. For comparison, *Wohlfahrt, Fenstermaker & Arnone (2008)* reported a net sink from 102 to 110 g C m$^{-2}$ year$^{-1}$ in the Mojave desert ecosystem. For a desert shrub community in Baja California/Mexico, the total annual carbon dioxide exchange was determined to range from −52 to −39 g C m$^{-2}$ year$^{-1}$ (*Hastings, Oechel & Muhlia-Melo, 2005*). In a riparian forest with similar mean annual temperature and precipitation, *Ma et al. (2017)* reported a net carbon sink from 278 g C m$^{-2}$ year$^{-1}$ to 427 g C m$^{-2}$ year$^{-1}$. Thus, the carbon sink capacity of the Tugai forest in the Ebinur Lake Basin is higher than that of the savanna in the Mojave Desert ecosystem of California and the desert shrub in Baja California/Mexico, while lower than the riparian forest.

The performances of the gap-filling methods in this study were comparatively good. It is difficult to determine the best applicability of the gap-filling methods in the desert Tugai forest via a single statistical parameter. Therefore, this study comprehensively evaluated the gap-filling models using $R^2$, RMSE, and BE. It was found that the test set $R^2$ obtained by the ANN method was 0.827. The test set $R^2$ obtained by *Menzer et al. (2015)* using the ANN method was 0.740, which was 8% lower than this study. The $R^2$ value obtained by the GP method was 0.844, which was 6% higher than that of *Menzer et al. (2015)*; $R^2$ obtained by the RBF method was 0.790, which was 39% higher than that of *Järvi et al. (2012)* and 3% higher than that of *Menzer et al. (2015)*. The RMSE of the test set obtained by the five methods ranged from 2.04 to 2.50 μmol m$^{-2}$ s$^{-1}$, which was smaller than that of *Järvi et al. (2012)* and *Menzer et al. (2015)*. This may be primarily due to the fact that the estimations by *Järvi et al. (2012)* and *Menzer et al. (2015)* were conducted on urban ecosystems in which the underlying surface of the observation sites was rougher, and the environmental factors affecting carbon dioxide exchange were more complex. Although the EC method requires that the underlying surface be uniform and flat, the position of the terrain is not absolutely guaranteed in the actual selection process of the underlying surface. In addition, when screening the modeling data, data with a signal quality of two were excluded (*Järvi et al., 2012*). In the current study, however, all data with a signal quality other than 0 (including 1 and 2) were excluded, resulting in high modeling data quality. It is clear that the selection of modeling data is particularly important when training models.

There were minor differences among the standard deviations of the annual sum of carbon dioxide exchange obtained by the five methods. Due to the low annual sum of arid desert ecosystem carbon dioxide exchange, its estimation is sensitive to data processing methods, with some methods possibly leading to bias (*Liu et al., 2010*). In addition, the soil volume water content data were distorted as a result of the high degree of soil salinization. Therefore, this study ignored the impact of soil moisture on carbon dioxide flux, leading to a very large difference in the annual sum of carbon dioxide exchange obtained by the five methods. It should be noted that this study was limited to the seasonal signal from only a single year's worth of data at the site. Future studies should utilize a longer dataset, which would provide more insight into the ecological functioning of the site.

The BE of the annual sum of carbon dioxide exchange obtained by the five methods ranged from −13.69 to 14.05 g C m$^{-2}$ year$^{-1}$. The minor differences found in the BE may be related to the fact that the carbon source/sink function of the Tugai forest is more affected by environmental factors than that of other ecosystems, especially soil moisture content. The impact of data processing methods on the estimation of the annual sum of carbon dioxide exchange was discovered to be highly variable, which may be related to the sensitivity of arid desert ecosystems to environmental factors, as well as the definition of valid data.

## Carbon dioxide exchange during nighttime and non-growing season daytime

This study used a quantile-based method to determine the outliers, and also identified and deleted the anomalies in the $F_C$ time series. After careful screening during data preprocessing, a total of 42 negative values of carbon dioxide exchange at night were observed, accounting for 1.9% of the nighttime data, which was far less than the 28% of temperate broadleaf forest observations (*Thomas et al., 2011*). The negative carbon flux observations indicate that there was carbon uptake in the underlying vegetation of the flux tower at night, although this phenomenon was unlikely (*Thomas et al., 2011*). There is currently no unified view to explain this phenomenon, ignoring atmospheric advection and precipitation as possible causes. Moreover, there is also a systematic error that stems from ignoring the atmospheric advection and precipitation measurements. Even though advection correction is attractive from a theoretical point of view, it is unlikely that air advection characteristics varied over the observed period and sites (*Feigenwinter et al., 2008*).

The accumulation of carbon dioxide exchange obtained by the five different methods performed consistently during the growing season and exhibited very large differences during the non-growing season (Fig. 10). During the growing season, the carbon flux data were relatively complete, with no long gaps; the carbon flux data during the non-growing season, however, had more gaps, and the continuously missing time windows were long (Fig. 5), leading to very large differences in method performance. The five methods displayed obvious carbon uptake characteristics in February (Fig. 10), which was consistent with the daily variation characteristics of the selected high-quality modeling data from January and February (Fig. 3A). This result may have been influenced by ignoring the advection and storage terms, which produced a certain systematic error (*Bowling et al., 2010*). In addition, the high saline alkali water and soil played an important role in the carbon uptake (*Li et al., 2015*; *Ma et al., 2013*; *Xie et al., 2009*).

To some extent, the valid data observed at the site depend on independent processes influenced by the geographic environment of the observatory. The resulting uncertainty is primarily random, depending on both the missing time window and the gap-filling method. The friction velocity $u^*$ is intended to capture the turbulent characteristics in the vicinity of the surface. Nighttime data under unsteady conditions are often removed by $u^*$ threshold screening. Each site has a specific $u^*$ threshold that is expected to vary over time (*Béziat, Ceschia & Dedieu, 2009*; *Moureaux et al., 2008*). The nighttime friction velocity

threshold (0.12 m/s) during the growing season was greater than the corresponding value for the non-growing season (0.10 m/s), indicating that the flux observation station experienced different air turbulence characteristics during different seasons (Fig. 6). A possible reason for this is simply that the air temperature during the growing season is higher than during the non-growing season. According to the turbulence intensity theorem, the turbulence intensity is a function of both velocity and temperature shears (*Hu, Chen & Zuo, 2007*). The near-surface temperature changes more rapidly during the growing season night than during the non-growing season night. Thus, in the near-surface layer, the temperature shear formed during the growing season night is larger than that formed during the non-growing season night. In addition, due to vegetation growth, the underlying surface roughness of the observatory is higher during the non-growing season than during the growing season (Fig. 6C). Uncertainty may still exist after $u^*$ threshold screening. There is no guarantee that the data screening process will eliminate all bad data, nor that only bad data will be removed.

## Uncertainty analysis

All measurements have errors or uncertainties (*Taylor, 1997*), and measurement uncertainty includes random and systematic uncertainties (ISO/IEC 2008). Systematic error is considered to be a constant but unknown deviation (*Abernethy, Benedict & Dowdell, 1985*) and must be estimated by empirical, theoretical, or supplementary measurements. Systematic error cannot be determined by statistical analysis of the measured data itself, nor can it be reduced by averaging. The bootstrapping method has been used to quantify the random uncertainty of the flux integral at different time scales (*Wang, Riley & Collins, 2015*). In addition, the Monte Carlo method is used to estimate the transmission of uncertainty over longer time scales, particularly the uncertainty associated with gap filling (*Moffat et al., 2007*).

Comparing the change in the empirical distribution of the residuals obtained by the five gap-filling methods (Fig. 9), it was discovered that the empirical distribution of these residuals approximated the Laplace (or double-exponential) distribution. Actually, the distribution of the $F_C$ random error from the Tugai forest ($|F_C| < 10$ µmol m$^{-2}$ s$^{-1}$) is leptokurtic. As a characteristic of random uncertainty, the leptokurtic distribution has been shown to be robust across a variety of sites and ecosystem types (*Hollinger & Richardson, 2005*; *Richardson et al., 2008*; *Stauch, Jarvis & Schulz, 2008*; *Lasslop et al., 2008*; *Liu et al., 2009*; *Wang, Riley & Collins, 2015*). It has been suggested that the leptokurtic distribution is the result of a superposition of Gaussian distributions and a non-constant variance (*Hollinger et al., 2004*; *Lasslop et al., 2008*). After normalizing the error (i.e., dividing each flux observation by the expected standard deviation), its distribution approximated a Gaussian distribution (*Richardson et al., 2008*).

We also found that the random uncertainty increased with increasing carbon flux (Table 5). This was consistent with the findings of *Richardson et al. (2008)* for the uncertainty of carbon flux measurements from model residuals. Using different methods to fill gaps leads to the differences among the distributions of the measured random errors and affects the estimation of the annual sum of carbon dioxide exchanges. Considering

the differences in site characteristics and types of ecosystems, as well as the degree to which the valid flux data are contaminated with data from a separate (nonbiological or atmospheric) process, it is necessary to evaluate random uncertainty carefully and site specifically. This study may provide a valid method for evaluating the gap-filling uncertainty of other types of ecosystems.

## CONCLUSIONS

This study estimated the annual sum of carbon dioxide exchange of the Tugai forest in the Ebinur Lake Basin of Northwest China in 2015. Selective system errors were corrected by determining the friction velocity threshold and utilizing it to extract high-quality data on the growing and non-growing seasons. Five different methods were used to fill the gaps in the carbon flux time series. These methods all exhibited higher $R^2$ and lower RMSE values than previous techniques. The GP method was the best of these gap-filling techniques. Uncertainty was inferred from the model residuals; the distribution of the residuals exhibited non-normal properties of spikes and thick tails. The ultimate goal of this investigation was not to select the best gap-filling method by comparing the merits of each but to estimate the uncertainty caused by the gap-filling and to correct model error. The total annual carbon exchange of the Tugai forest at the observation site in 2015 was $-163.50 \pm 9.43$ g C m$^{-2}$ year$^{-1}$, indicating that the Tugai forest in the Ebinur Lake Basin is a carbon sink. The errors from different sources were complex, making it difficult to give accurate estimates; therefore, this study only estimated the random errors and bias. Estimating errors from different sources will be the focus of uncertainty studies on the cumulants of carbon dioxide exchange in the future. Exploring the function of the Tugai forest in arid regions still requires additional understanding of the underlying ecological mechanisms.

## ACKNOWLEDGEMENTS

We thank LetPub for its linguistic assistance during the preparation of this manuscript.

### Funding

This work was supported by the National Natural Science Foundation of China (NO. 31560131). The funders had no role in study design, data collection and analysis, decision to publish, or preparation of the manuscript.

### Grant Disclosures

The following grant information was disclosed by the authors:
National Natural Science Foundation of China: 31560131.

### Competing Interests

The authors declare that they have no competing interests.

## Author Contributions

- Dexiong Teng performed the experiments, analyzed the data, prepared figures and/or tables, authored or reviewed drafts of the paper, and approved the final draft.
- Xuemin He performed the experiments, authored or reviewed drafts of the paper, and approved the final draft.
- Jingzhe Wang conceived and designed the experiments, prepared figures and/or tables, and approved the final draft.
- Jinlong Wang performed the experiments, prepared figures and/or tables, and approved the final draft.
- Guanghui Lv conceived and designed the experiments, authored or reviewed drafts of the paper, and approved the final draft.

## Field Study Permissions

The following information was supplied relating to field study approvals (i.e., approving body and any reference numbers):

The Xinjiang Ebinur Lake Wetland National Natural Protection Zone Management Bureau approved site access.

## Data Availability

The raw measurements are available in the Supplemental File. The code in the paper comes from online open source projects (https://cran.r-project.org/).

## Supplemental Information

Supplemental information for this article can be found online at http://dx.doi.org/10.7717/peerj.8530#supplemental-information.

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
