# Peer review of "Uncertainty in gap filling and estimating the annual sum of carbon dioxide exchange for the desert Tugai forest, Ebinur Lake Basin, Northwest China"

_PeerJ, doi:10.7717/peerj.8530_

## Round 0.1 · original submission · Major Revisions

Given the detailed comments of both the reviewers I would encourage you to deal with each of the suggestions in full. I would particularly welcome more detail on the site characteristics and vegetation types (see reviewer two's comments on this). A fuller description of the limitations and corrections of the methods of EC (at the site and in general) should be made including the corrections for frequency loss, storage terms etc..

Reviewer 1 ·

Basic reporting

The manuscript “Uncertainty in gap-filling and estimating the annual sum of carbon dioxide exchange for the desert Tugai forest, Ebinur Lake Basin, Northwest China” applied four different gap-filling methods to evaluate uncertainties in annual NEE over a desert forest in northwest China, to provide a robust flux estimate obtained from the eddy covariance measurements. In my opinion, the authors have done a good work (grammar in general, figures, and tables) but I still think the manuscript could be further improved and more clearly written.

Experimental design

Clearly described.

Validity of the findings

Please see more comments below.

Additional comments

General comments:
1. Why did you choose these four particular gap-filling methods rather than others (e.g. the MDS method widely used in EuroFlux and ICOS flux networks) to assess the uncertainty? Do you have a reason for your preference? Please emphasize this more in your introduction.
2. It would be great to stress more on how important this particular work is. For example, would the gap-filling uncertainty evaluation only apply for the desert forest? Or how applicable it will be at other types of ecosystems?
3. Is the random error distribution applicable to other sites/ecosystems?
4. Would it be possible to perform a simple marginal distribution sampling method (available at https://www.bgc-jena.mpg.de/bgi/index.php/Services/REddyProcWeb) to see how off/close this standard gap-filling method could be, compared to the four methods used in this manuscript.
5. Language may need a bit more work. There are several places that are not clearly written or even misleading, please refer to the detailed comments and make changes where necessary.

Detailed comments:
Line 130: Suggest to add more details on gap distribution and attribution, e.g. are there any long gaps? How many “very short gaps”, “short gaps”, “medium gaps”, “long gaps” (e.g. refer to Moffat et al. 2007)?
Line 170: suggest to cite Papale et al. 2006 rather than Liu et al. 2010.
Line 188: suggest to change “The model” to “The gap-filling model”.
Line 194: It should be Papale and Valentini 2003.
Line 202: Please change it to “…by Schmidt et al. 2007”.
Line 208-209: It should be Gelman 2006, rather than Gelman and Andrew…
Line 271: what do you mean by “effective data points”?
Line 274: what’s the “effective” ratio? Please define.
Line 287: what do you mean by “theoretical and practical deficiencies” here? Please be specific.
Line 288: “…is filtered…”, not “filtering”
Line 290-291: awkward sentence. Suggest to rephrase it like “222 and 192 nighttime Fc data were eliminated by the u* threshold during the growing and non-growing seasons, respectively.”
Line 292-294: Did you remove the negative nighttime flux data? Why or why not?
Line 300: why “the best results”? Did you get other different results than the best ones?
Line 301-302: suggest to delete “with a difference of about 6%”, it’s so obvious that you don’t have to repeat here.
Line 314-316: This sentence is not clear, please rephrase it.
Line 319: how did you quantify the changes in the residuals and compared among the four methods?
Line 335-337: awkward sentence. Please reword it.
Line 342-348: Isn’t this part of methods?
Line 365-367: Reference? Is it assumed by others, or your own assumption?
Line 373: please delete the parentheses on “standard deviation”.
Line 387: “potential carbon sink in the most unlikely place on Earth”, please rephrase it in a logic way.
Line 388: Are you talking about the global missing carbon sink?
Line 387-392: This part reads more like introduction and motivation of your study, which is not recommended to be listed in discussion.
Line 396-400: what is point of comparing your study to a spruce forest?
Line 401-403: which model are you talking about here?
Line 403-407: Please reword it in a concise way, like “the R2 values obtained in this study are all xx% higher than the previous work (refs).”
Line 409-410: Suggest to move this sentence “Thus, …good.” somewhere at the beginning of this paragraph.
Line 451: please change “absorption” to “uptake”
Line 460: why this paragraph is separated from above?
Line 464: Are you indicating the thermal generated mechanism that was reflected by the air temperature difference? If so, please add one or two sentences to discuss more.
Line 501-502: You can’t make the conclusion that the four methods resulted in higher r2 and smaller rmse than previous techniques when you didn’t use any previous technique at your site, unless if you could prove that these gap-filling models are not site dependent. So please reword it.
Line 503-506: Are you in favor of using different gap-filling methods to estimate the uncertainty? Isn’t the random error distribution at your site another more important message that everybody could use for their desert forest sites in the future? Suggest to add more conclusive message that readers could benefit the most from your study.

Figure 1b, It looks like the tower is located on the lake, please modify the red dot. Please add a scale for figure 1c.
Figure 2 caption, FC, “C” should be subscripted. “…over two-month periods”
Figure 4: the asterisk of ustar on the y axis should be subscripted.
Figure 5: CO2 in the flux unit is wrongly written in the figure caption, please correct it. Suggest to label the name of gap-filling model in each figure panel too.
Table 4: It would be very nice to add the sums for growing and non-growing seasons to this table.

·

Basic reporting

Overall, the research is reported clearly and the ms indicates an appropriate and thorough approach to data collection and data analysis, with clear presentation of the core results. The results are of substantial interest to the global terrestrial carbon cycle community, and the relevant literature is well referred to. One area that requires more information is over the site description (see below). However, the English language in several places needs improvement in order to encourage a wider readership, with some repetition, and unusual choice of words and in the clarity of structure and logic. As an example of the latter, the introduction and rationale explained for the paper is not very strong – the eddy covariance method has been used for several decades (not ‘few’, and not ‘gradually become’) and this has been due not just to the development of robust sonic anemometers, but because of robust gas analysers (not mentioned). ‘covariance of wind pulsation and physical pulsation’ is a strange phrase, ‘ covariance of fluctuations in vertical wind speed and gas concentration’ is a more normal way to describe the basis of the approach. The list of ecosystems measured is OK, but a single review paper (such as the Baldocchi one referred to earlier) would be a better reference than singling out particular examples from the hundreds available. The first paragraph ends by saying ‘few constant (continuous ?) observations of energy and material fluxes in arid desert ecosystems’ – which is true, but this paper doesn’t report energy balances only CO2 fluxes, so sets the reader with the wrong expectation. The 2nd para lists various factors that result in ‘data loss or abnormalities’, but omits the usually more frequent cause which is when the methodological assumptions are not met – e.g. limited turbulence, non-stationarity, and technical issues such as instrument resolution, frequency response. As the paper is presenting some novel data on the C balance of the Tugai forest ecosystem, it would have been useful to have some info on the extent of this habitat in the introduction to strengthen the case for the importance of these results.

Experimental design

The core eddy covariance methods and instruments and calculations described are very much standard in the C flux measurement field, appropriate and well carried out. However, more information on the ecosystem and site would be most helpful to readers, as most will be unfamiliar, and I found the information given rather confusing For example, it was surprising that this study in a ‘continental arid climate zone’ is actually in a ‘wetland reserve’ with the measurement site ‘on the bank of a river’ or ‘near the A. River… in a desert ecosystem influenced by lakes and rivers’. It is also described as an ‘estuary of strong winds’, which is not clear. More info on the actual site and area being measured would be helpful, and necessary to understand and interpret the results. Is there a previous accessible publication from the project that could be referred to for details? ‘The dominant plants in this region…’ are given but does that mean the veg of the site and thus the vegetation CO2 flux being measured ? If Phragmites is on the site, then I am surprised that this is ‘arid’ and this vegetation would seem more like swamp or riverine, possibly seasonal. Similarly, doesn’t presence of Populus euphratica suggest a phreatic vegetation type ? The text only says that there are two ecosystems wetland and arid grassland-forest – was the site completely in the grassland-forest? Also it would be helpful to explain how much of the vegetation is evergreen – ( I looked up Haloxylon which is, but presumably the poplar isn’t) – to understand the seasonal CO2 flux patterns. The aerial photo of the site (1c, taken when in the year ? - a larger one of these would be more useful) does not seem to show Phragmites, and it doesn’t look like 60% vegetation cover, although it is very easy to be misled by a photo. It would be helpful to know the seasonal pattern of vegetation cover as it probably varies greatly given the climate pattern. To understand the CO2 flux seasonality pattern shown in Fig 2 I suggest a first graph of the weather conditions during the year of measurements would be very useful – air temp, radiation, precip, wind speed. Nothing is said about the soil type(s) present at the site. Also, I note that the non-growing season is described as ‘Jan-April’ which seems to conflict with Fig 3 which shows substantial CO2 uptake in March/April, and some even in Jan/feb.
The main gaps in the analysis description is that no mention is made of any frequency loss corrections made (but should be small for an open path set-up); the storage flux term (probably small given the vegetation type and height – but should be stated, i.e. that they assume NEE = Fc, they do state 'total CO2 flux..' on l. 116 ) nor of the site fetch (the extent of the homogeneous area around the tower) nor of any footprint estimates or filtering for wind direction or filtering. If the site is near a river, then would it affect the flux measurements ?
NDVI information is used (l. 135) but no information about the data source.

Validity of the findings

Potentially a useful set of CO2 flux results for a novel or at least under-reported ecosystem type.
The authors have been open about the limitation of not having soil water content data to help explain the observed patterns, (around l. 425) which in an arid zone site is indeed a pity. Another limitation is that this is just one year of results, and interannual variation could be large; although this does not undermine the work it is not stated clearly or discussed. Neither are the conclusions clear about which gap-filling method is 'the best' for this data set..

Additional comments

The information on data set size, and how much ‘lost’ due to different problems and filtering could all be done more concisely and easily understood with a table.
Fig 2. – need to state if these mean Fc diurnal patterns are before/after u* or other filtering.
The stats in figure 4 for night-time u* don’t agree with the statement l. 284 about which months are sig diff from others. Is this graph actually informative as it depends on wind speed seasonality as well as vegetation characteristics ? a graph of u* threshold determinations might be more useful ? While establishing u* threshold is important step, is the estimated difference in u* threshold between seasons (0.10 and 0.12) significant ? The statement in the conclusion (l. 499) that ‘this approach prevented the u* from being affected by the roughness of the underlyng surface and vegetation’ is hard to understand and probably wrong. Is it trying to say that ‘Fc values that might have been affected by low u* were filtered out’? – but this is hardly a conclusion and a standard practice anyway.
Eng lang notes: ‘friction velocity’ not ‘friction vorticity’; ‘outliers’ not ‘wild spots’ l. 434. not clear what ‘mutation points’ means l.435? Not usual to say ‘enormous differences’ but ‘very large’ (l. 383 , 447, 450.
Fig 5. Graph b with the GP method seems to represent the full range of scatter in the observations much better – the others all show upper and lower ‘limits’ or ‘saturation’ suggesting strong underestimation of large (in absolute sense) fluxes ? this doesn’t seem to agree with the comment that stats parameters all ‘very similar’ l. 303?.
Fig 6 – l. 305 says the gap filled Fc is more obvious diurnal patter’ – than what ? the un-filled – but not shown? Not easy to see differences between these plots.
The cumulative plots in Fig 8 are rather surprising, showing net CO2 uptake even from February, for 3 of the methods at least.

The conclusion say that the aim was to provide a ‘reliable method’ for interpolation (l. 505) – but I was not left with your recommendation or conclusion of which method of those 4 examined is reliable ?
In the discussion it is important to note that the annual C balance established in this novel data set s just for one year – important given the inter-annual variation often observed in arid environments – particular with rainfall. Is it possible to compare the rainfall measured in 2015 with a longer term record for the site ?

The discussion compares C balance results here with the Californian oak/grass savanna, which seems a useful comparison, but also with a temperate spruce forest which seems an odd one to pick of the many forest sites. (l. 400 or so).

---

## Round 0.2 · Minor Revisions

I would recommend following the reviewer's comments closely and giving some space to the interpretation of the particular aspects of the data set in addition to the comparison of the gap filling techniques.

·

Basic reporting

Improved language - see comments below re Fig 1 clarity

Experimental design

Interesting and unique data set and thorough analysis, but see comments below about clarifying what areas have been measured.

Validity of the findings

Thorough analysis, unique data set. Would benefit from more comment about basic results prior to the analysis and gap-filling.

Additional comments

This ms has been improved during the revision. Key changes have been inclusion of the ‘standard’ gap-filling MDS method among the methods tested, more information about the site including soils and weather conditions (Figs 1 & 2) and the English language has been improved.
Some issues that were noted before, now stand out more clearly, however. Chief of these is that they have included a ‘footprint’ graph (in Fig 1, although it is very small and not very clea, the labelling on the contours is illegible – I suggest this paper merits a clear aerial photo or satellite image of the site and the footprint separately – the previous version showed the tower and the vegetation, although this has been replaced by the footprint now). Fig 1 c seems to show that the footprint covers both the river/green areas (is this a satellite image ? ) and the desert. I raised the question in the previous version – are the measured fluxes typical of a desert or of a riverine forest that they now mention (L. 113 “in the forest-desert transitional zone influenced by lakes and rivers”)? Is it “a desert-oasis ecotone” that they mention in their response to reviewers ? Where they also mention “flux measurements influenced by the river is inevitable” – but this is not mentioned in the paper, which discusses the results as for a desert. Did they find any differences in fluxes when they looked at different directions or wind sectors ? (or was the wind direction very seasonal which makes such interpretation difficult ? ). If the river or riverine vegetation was in the footprint, it might affect fluxes as water can obviously be a source or sink of CO2 depending on chemistry and temperature etc. Clarifying what areas the fluxes were from would help readers wanting to understand desert C balances – which is stated as one of the aims of the work in the intro.
However, it seems from the way the paper is focussed that the real aim is to compare different gap filling methods, and not to actually interpret or comment on the C balance info, which seems to be missing an important contribution of their unique data set. For example, no comment is made about the measured C uptake during the daytime in Jan-Feb – despite the below zero temperatures and the low radiation at that time. Secondly, while they examine with statistical parameters the different gap filling methods, they don’t comment on their overall ability to follow the seasonal cycle as shown in Fig 10. Only HLM and MDS method seems to have an appropriate low/zero C accumulation in the winter, whereas the other methods have substantial uptake. I would suggest that the MDS pattern seems more ‘believable’ - but the authors know the site and the data they have, and should comment. The ANN method in particular shows a pronounced uptake in Feb – possibly coinciding with the Feb gap period ? I also note that the final cumulated C values in this Fig 10 do not seem to agree with Table 7 values – in the figure HLM and RBF are similar, MDS the smallest, and GP and ANN the lowest.
Minor editorial points:
NDVI data source is only mentioned in a table and nothing about what spatial resolution and what frequency.
L. 263 repeats L. 248

---

## Round 0.3 · Minor Revisions

Thanks you very much for your engagement with the reviewer who kindly reviewed your redrafted manuscript. I would be pleased to accept the manuscript for publication on completion of some very minor changes.

The reviewer highlighted some uncertainty with the data being used to in table 7 and figure 10 which you had responded to in your response, however I would like to see some clarification in the text and table/figure captions as to the difference between the data used to derive the budgets.

---

## Round 0.4 · accepted · Accept

Thank you responding positively to the comments of the reviewers and for your patience in going through the review process. We appreciate your support of the journal and hope you will publish future work with PeerJ.